# The Case for Comorbid Myofascial Pain—A Qualitative Review

**DOI:** 10.3390/ijerph17145188

**Published:** 2020-07-17

**Authors:** Simon Vulfsons, Amir Minerbi

**Affiliations:** Institute for Pain Medicine, Rambam Health Care Campus, Haifa, Israel Rappaport School of Medicine, Technion Institute of Technology, Haifa 31096, Israel; a_minerbi@rambam.health.gov.il

**Keywords:** myofascial pain syndrome, comorbid pain, chronic pain, secondary pain

## Abstract

Myofascial pain syndrome is widely considered to be among the most prevalent pain conditions, both in the community and in specialized pain clinics. While myofascial pain often arises in otherwise healthy individuals, evidence is mounting that its prevalence may be even higher in individuals with various comorbidities. Comorbid myofascial pain has been observed in a wide variety of medical conditions, including malignant tumors, osteoarthritis, neurological conditions, and mental health conditions. Here, we review the evidence of comorbid myofascial pain and discuss the diagnostic and therapeutic implications of its recognition.

## 1. Introduction

Myofascial pain syndrome (MPS) is a regional pain syndrome arising in muscles and muscle fascia, characterized by tenderness to palpation, limited range of motion and the presence of taut bands [1,2,3]. The diagnosis of MPS rests strictly on clinical criteria, including a history and physical examination of the involved muscles [4,5,6].

Myofascial trigger points (MTrPs) are considered a cornerstone of the pathophysiology of the MPS by many authors [3,7,8,9,10]. MTrPs are defined as hypersensitive, palpable, taut bands of muscle that are painful to palpation, reproduce the patient’s symptoms, and cause referred pain [3,11]. They are typically divided into active MTrPs, which reproduce the patient’s familiar local and referred symptoms and latent MTrPs, which elicit other, unfamiliar symptoms [3]. While the presence of MTRrPs is considered by many authors to be a key characteristic of MPS, their identification and significance are a matter of ongoing debate [1,12,13]. Some authors have disputed the existence of myofascial trigger points [13]. As the controversy surrounding the significance of MPS is beyond the scope of this review, here, we focus on the clinical syndrome of MPS.

The prevalence of MPS is also a matter of debate in light of the lack of widely accepted diagnostic criteria and the different settings in which the syndrome has been evaluated [5,6,14,15,16,17,18]. Nevertheless it is generally appreciated that MPS is a common entity, responsible for many primary care visits and specialist consultations [1,15,19]. As appreciation of the syndrome increases, clinicians from different fields of medicine diagnose it in individuals with a wide range of medical conditions [20]. Despite the fact that MPS often arises in otherwise healthy individuals [1], evidence is mounting that in individuals with various comorbidities, its prevalence may be even higher, and these observations will be reviewed here in detail.

MPS has been reported in patients with a wide variety of regional pain syndromes, such as frozen shoulder [21], whiplash associated pain [22], lower back pain [23], anterior knee pain [24], and many others. In these clinical syndromes, pain may be generated by various processes, making MPS one of the possible etiologies to consider in the differential diagnosis.

In contrast, comorbid MPS should be considered when MPS is diagnosed in a patient with another independent diagnosis, such as malignant, rheumatologic, neurological, or psychiatric diseases, to name just a few. In these cases, MPS should be considered not as a possible etiology of the primary diagnosis but rather as a comorbid condition, possibly contributing to the symptoms of the patient.

As pain is a common symptom of many medical conditions [25,26,27], it is important for the clinician to differentiate pain that is generated by the primary disease from pain which originates in comorbid processes such as MPS. This distinction may allow the clinician to utilize appropriate therapeutic modalities to treat the patient’s pain [1]. In this article, we review the prevalence of MPS accompanying various medical conditions, painful or otherwise, and discuss the clinical pertinence of this entity.

## 2. Methods

This review is based on a literature search performed between 20 and 31 March 2020, using PubMed database. The search included articles containing a combination of (1) key words pertaining to MPS and (2) key words pertaining to potential comorbid conditions. The following key words were used to select for MPS: myofascial, myalgia, muscle pain, torticollis. The following key words were used to select for potential comorbid conditions: (1) primary headaches—headache, migraine, occipital neuralgia, cluster headache, tension headache; (2) oncologic diseases—malignant, cancer, tumor, oncologic, chemotherapy, radiotherapy; (3) arthritis—arthrosis, osteoarthritis, spondyloarthritis, spondyloarthropathy; (4) infectious diseases—infection, inflammation, bacterial, viral, meningitis, sinusitis, otitis, cellulitis, tonsillitis, urinary tract infection; (5) neurological diseases and chronic pain conditions—complex regional pain syndrome, neuropathy, neuropathic pain, radicular pain, radiculopathy, Parkinson’s disease, stroke, cerebrovascular accident, spinal cord injury, dementia; (6) mental health and insomnia—anxiety, depression, stress, mental health, post-traumatic stress disorder, insomnia.

Of the resulting articles, non-English manuscripts, reviews, case reports, and animal studies were excluded.

### 2.1. MPS Accompanying Oncologic Diseases

Pain is a common symptom among individuals with cancer. It has been estimated that 38% of cancer patients at all stages of disease report pain [28]. Frequency of pain among patients undergoing active oncological treatment is higher (55%), as is pain frequency among patients with advanced disease (66%). Cancer survivors also report a higher than average pain prevalence, estimated at 38% [28]. Pain in cancer patients can be attributed to the primary oncological disease, to anti-cancer treatments (chemotherapy, radiotherapy, surgery, biologic, and hormonal therapy), or to unrelated factors [29]. The contribution of MPS to cancer pain has been observed in several studies, most often in the context of breast cancer, head and neck tumors, and advanced cancer.

MPS in patients with head and neck cancer has been described for several decades [30,31]. Chua et al. retrospectively evaluated the charts of 40 consecutive patients with head and neck cancer who had completed therapy and had no evidence of disease [32]. Of these patients, 52% had severe pain, of which 13% were deemed to have MPS. Similar results were reported by Cardoso et al. in a prospective study, including 167 patients with head and neck cancer after a disease-free interval of a year or more. A total of 57% of patients reported significant pain, of which 11.9% was MPS, most often involving the upper trapezius muscle [33]. Recently, Ortiz-Comino et al. compared the prevalence of trigger points in a case control study of 30 individuals with head and neck cancer suffering from cervical and/or temporomandibular pain with that of 30 matched pain-free controls. Active trigger points (defined as points that elicit the patients’ typical pain when pressed) were found only in the patient group [34]. The preselection of cancer patients with pain as opposed to healthy controls with no pain is a significant limitation of this study and indeed of many studies addressing this topic.

The prevalence of MPS in women with breast cancer has been addressed in several studies. Torres Lacomba et al. prospectively followed 116 women with breast cancer who were undergoing surgery with axillary lymph node dissection for the duration of one year [35]. MPS developed in 44.8% of the patients and was the most common cause of pain in this cohort. The peak onset of MPS was approximately six months post-surgery, and the muscles that were most commonly involved were latissimus dorsi, serratus anterior, pectoralis major, and infraspinatus. Ko et al. retrospectively reviewed the charts of 52 patients with breast cancer who had undergone both surgery and chemotherapy [36]. The prevalence of MPS among these patients exceeded 20%, of which 13.5% had isolated MPS, and 7.5% had combined MPS and chemotherapy-induced neuropathic pain. The risk of developing MPS was increased with longer duration of chemotherapy treatment and with hormone treatment. Shoulder and chest muscles were most commonly involved. Fernández-Lao et al. compared the frequency of shoulder and neck pain among breast cancer patients undergoing either lumpectomy, mastectomy, or no surgery [37]. Myofascial trigger points were more common after surgery, but no significant differences were observed between patients undergoing lumpectomy and those undergoing mastectomy. Two other prospective studies reported a decreased pressure pain threshold and increased frequency of active trigger points in the cervical and shoulder muscles of breast cancer patients with neck and shoulder pain as compared to a matched group of healthy controls with no pain [38,39]. As noted above, these results should be viewed in light of the selection bias of patients with pain in the cancer group as opposed to pain-free patients in the control group.

The prevalence of MPS among patients with advanced cancer, receiving palliative care, was assessed by Ishiki et al. [40]. Thirty-four patients with variable malignant diseases, who were deemed incurable, were included in this retrospective study. The most prevalent primary diagnoses were pancreas, breast, lung, colorectal, and head and neck tumors. Pain was reported by 65% of the patients, of which 45% were diagnosed with MPS according to the Rivers criteria and 90% according to the Simons criteria [6,17]. 

The efficacy of myofascial trigger point injections in patients with advanced cancer was evaluated in two studies, demonstrating decreased spontaneous and pressure-evoked pain [41,42].

In conclusion, while the accurate prevalence of MPS in patients with cancer is difficult to determine, pain is a common symptom in this population, and MPS would appear to contribute substantially. The recognition and treatment of this component should be sought in all cancer pain patients.

### 2.2. MPS Accompanying Osteoarthritis

Pain is a common symptom of osteoarthritis (OA) [43]. While pain arising in the joint is widely recognized in patients with OA, there is a growing appreciation of the contribution of the surrounding myofascial tissues to pain in these patients. Bajaj et al. explored the prevalence of myofascial trigger points in a group of 14 patients with OA of the hip, knee, or both as compared to that of a matched group of healthy controls [44]. A significantly higher number of trigger points was identified in patients as compared to controls, and a positive correlation was observed between the number of recognized trigger points and OA radiological scores. Henry et al. explored the prevalence of MPS in a cohort of 25 patients on a waitlist for total knee arthroplasty due to OA of the knee [45]. All patients were found to have MPS of the muscles surrounding the knee, with the gastrocnemius being most commonly involved. A significant improvement in pain score and function was observed following the injection of bupivacaine into these muscles. In a case-control study, Albuquerque-García et al. compared the number of active trigger points in the muscles surrounding the knees of 18 women with bilateral painful knee OA to those of 18 matched controls [46]. A higher number of myofascial trigger points was detected in OA patients, correlating with a higher intensity of pain and lower functional capacity. In a cross-sectional study by Sánchez-Romero et al., 114 patients with OA of the knee were evaluated for myofascial trigger points [47]. Active myofascial trigger points were detected in 75% of participants’ vastus medialis muscles and in 65% of their vastus lateralis muscles. Finally, several studies have explored the clinical efficacy of treating the myofascial component of OA-related pain, mostly showing short-term favorable results in pain and function [45,48,49]. In a randomized controlled trial including 40 patients undergoing total knee replacement, dry needling of the hip and calf muscles under anesthesia resulted in decreased post-surgical analgesic consumption, as well as decreased pain intensity one month post-surgery, when compared to sham needling [50].

Taken together, these studies suggest that MPS may contribute to overall pain in individuals with OA and that treatment of this component may confer a short-term clinical benefit to patients.

### 2.3. MPS Accompanying Neurological Diseases and Pain Syndromes

MPS has been described in several neurological conditions associated with pain. Stroke is a common cause of chronic pain, either central or secondary to paralysis and spasticity. In a cross-sectional study of 50 post-stroke individuals with shoulder pain, active myofascial trigger points were documented in up to 50% of the examined infraspinatus muscles [51], while other muscles were less often affected by MPS (supraspinatus 34%, teres minor 12%, and upper trapezius 20%). Weakness, rigidity, and adhesive capsulitis have all been implicated in the development of post-stroke shoulder pain [52], often accompanied by MPS of the subscapularis and other shoulder girdle muscles [53,54]. In another observational study, de Oliveira et al. examined 40 patients with central post-stroke pain, observing MPS in 67.5% of them [55]. Several studies explored the clinical efficacy of treating the myofascial component of post-stroke pain, demonstrating reduced pain intensity and improved range of motion [56] as well as improved balance and function [57].

In patients with complex regional pain syndrome (CRPS), proximal myofascial involvement has been reported by several authors: for example, Allen et al. conducted a cross-sectional study of 134 individuals with CRPS, reporting that 56% of them had a myofascial component to their pain [58]. In a retrospective chart review of 41 CRPS patients, Rashik et al. found myofascial dysfunction of the involved limb in 61% of the patients, more often in upper extremity CRPS (70%) than in lower (47%) [59]. Recently, Dor et al. compared the presence of thoracic paraspinal myofascial trigger points in a case-control study involving 20 individuals with upper extremity CRPS and 20 healthy controls [60]. The prevalence of active myofascial trigger points in the different muscles which were evaluated was 15–35% in CRPS patients, as opposed to 0% in the control group. Finally, several cases and case series were reported in which the treatment of comorbid MPS led to significant symptomatic relief in patients with CRPS [61,62,63].

Lumbosacral radicular pain was associated with a significantly higher prevalence of gluteal myofascial trigger points (76% vs. 1.9%) in a prospective case-control study performed with 271 patients with lumbosacral radicular pain and 152 healthy individuals [64].

Similarly, a high prevalence of MPS was observed in individuals with spinal cord injury [65], brachial plexus injury [66], trigeminal neuralgia [67], diabetic peripheral neuropathy [68], and post-thoracotomy pain syndrome [69].

Overall, these studies suggest that MPS is a common comorbidity accompanying multiple neuropathic, central, and other neurological pain conditions.

### 2.4. MPS Accompanying Primary Headaches

MPS associated with headaches has been well documented for more than 60 years. In his summary in 1957, Bonica described several MPS syndromes, including in the head and neck region, causing chronic headaches [70]. Other authors from this period have also described an association between MPS and chronic pain in the head and neck region [3,71]. More recently, Fricton describes a high prevalence of MPS in various head and neck pain syndromes. Thus, in a study of 296 patients evaluated for head and neck pain, 164 (55.4%) exhibited trigger points attributable to their pain. The patients complained of a wide variety of pain sites, including supraorbital, forehead, temple, post-auricular, vertex, occipital, and retro-orbital [72]. Other studies have shown high myofascial involvement, such as pain due to masticatory muscle MPS [73] and temporomandibular joint dysfunction syndrome [74].

Cervicogenic headaches, as described by Sjaastad [75], have been associated with an increased prevalence of MPS on the affected side [76]. Successfully treated cases of patients with chronic headaches lasting many years that were eventually attributed to MPS have been described [77]. Indeed, biomechanical factors involving neck musculature and posture have been attributed to the etiology and maintenance of chronic headache in patients suffering from this condition [78,79,80,81].

Migraine sufferers have also been found to demonstrate an increased prevalence of MPS, especially on the affected side. Tfelt-Hansen et al. described 50 migraine sufferers, of whom all but two had tenderness in the head and neck muscles. Of these, 73% also demonstrated referred pain characteristics. Injections of 1 mL lidocaine 1.5% or 1 mL saline into these tender points abolished the pain in 26 of 48 patients [82]. A later study designed to assess the efficacy of head and neck trigger point treatment in migraineurs found that treatment with local anesthetic infiltration decreased cutaneous and subcutaneous pain thresholds and decreased migraine pain intensity in a treatment group of 54 patients [83]. In another study, neck pain was found to be highly prevalent in patients with migraine with or without tension type headache, and myofascial tenderness was significantly increased in patients with neck pain [84]. Fernández-de-Las-Peñas et al. found that migraine subjects showed a significantly greater number of active, but not latent, trigger points when compared to healthy controls. Involved muscles were mostly located ipsilateral to the migraine headaches [85].

Patients suffering from tension type headaches (TTH), both episodic and chronic, have increased incidence of myofascial trigger points in their neck musculature [86,87,88]. In these studies, the referred pain elicited by active trigger points in the neck and shoulder muscles reproduced the headache pattern in patients with frequent episodic and chronic TTH.

Thus, it is clear that in the common forms of primary headache and cervicalgia, MPS is commonly associated and can be considered comorbid.

### 2.5. MPS Accompanying Inflammatory and Infectious Diseases

In comparison to other putative triggers of comorbid MPS, focal infectious and inflammatory conditions have been less frequently described in the literature. The potential of localized inflammatory and infectious processes to cause muscle irritation, leading to contraction, stiffness, and pain, has been widely documented in multiple clinical syndromes, such as nuchal rigidity accompanying meningitis [89], torticollis accompanying head and neck infections [90], and abdominal rigidity and guarding associated with intra-abdominal pathologies [91]. Nevertheless, even though the involved muscles are typically contracted, tender, and of limited range of motion, whether these conditions should be viewed as secondary localized MPS is debatable. Several studies provide hints, suggesting that localized MPS may accompany some inflammatory and infectious diseases. In an early case series, Cohen et al. reported on five cases of trismus and MPS, two of which proved to be secondary to infection and three secondary to cancer [30]. In a recent prospective study, Niraj et al. followed 54 patients with abdominal pain due to chronic pancreatitis [92]. Twenty-one (38%) patients had abdominal wall MPS which, in almost all cases, responded to a transversus abdominis plane block. In another prospective study by Niraj et al., 120 individuals with abdominal wall MPS were followed, of which 12.5% had ongoing visceral inflammatory processes that were possibly irritating the abdominal wall musculature [93]. Pelvic pain and dysuria are commonly associated with MPS [94]. In a retrospective study of 250 women presenting with symptoms of urinary tract infection, 50% were found to have pelvic MPS, but only 6% had positive urine cultures [95]. This observation suggests that irritative urinary symptoms may be associated with MPS and that, in some cases, pelvic MPS can accompany urinary tract infections.

Other studies have found an association between MPS of the abdominal wall and chronic visceral urogenital disease. These disorders include endometriosis and chronic nonmalignant pelvic pain [96,97]. Indeed, it seems evident that visceral disease can produce somatic muscle pain and myofascial trigger points.

It thus appears that while infectious and inflammatory processes have the potential to irritate muscles and to cause stiffness and tenderness, the evidence of MPS secondary to infectious and inflammatory pathologies is still limited.

### 2.6. MPS in Mental Health Conditions and Insomnia

The association between chronic pain and mental health conditions is well established and is widely believed to be bidirectional [88,89,90]. The prevalence of depression and anxiety is increased in individuals with chronic pain, and the prevalence of pain is increased in individuals with affective disorders and PTSD [98,99,100]. In this regard, it should come as no surprise that the prevalence of depression and anxiety is also increased in patients with MPS—individuals with MPS in different body regions showed a higher prevalence and severity of depression [101,102,103,104] and anxiety [105,106,107] when compared to healthy controls. This association raises two questions: (1) is the prevalence of anxiety and depression in MPS similar to that which is observed in other chronic pain conditions? (2) What is the direction of causality: could mental health conditions be risk factors for the development of MPS? Could MPS predispose to the development of mental health problems, or could the correlation be bidirectional?

Giannakopoulos et al. evaluated the prevalence of depression among a cohort of 242 individuals with temporomandibular area pain. Participants were divided by the origin of their pain into MPS, joint, or mixed origin. Patients with exclusive MPS were significantly more likely to have depression as compared to patients with exclusive joint pain [108]. In another cohort of 649 patients with facial pain, Mongini et al. divided participants by the etiology of their pain to myogenous, joint, neuropathic, and facial pain disorders [109]. Anxiety was most prevalent among patients with myogenic fascial pain, and the severity of anxiety and depression was independently correlated with the likelihood of having muscle tender points in all groups. In contrast, Maslak-Beres et al. found increased depression and stress levels among 260 patients with temporomandibular region pain, regardless of the etiology [110]. Faucett et al. compared the prevalence of depression among patients seen at a rheumatology clinic suffering from either MPS (67 patients) or arthritis (83 patients) [111]. Results indicated that MPS patients suffered from more pain and higher depression levels than arthritis patients. Interestingly, patients with MPS also exhibited less social support and increased conflict about their pain than arthritis patients. Taken together, these data suggest that the prevalence of affective disorders is increased in patients with MPS in comparison to other chronic pain conditions.

Vidaković et al. evaluated the prevalence of MPS in a cohort of 101 Croatian war veterans with PTSD and depression [112]. They found MPS of the upper body in 58% of the patients, indicating that the prevalence of MPS is significantly increased in this population. Vedolin et al. followed 35 young students during the academic year and found higher anxiety levels and elevated muscle tenderness during examination periods, regardless of the presence of previous pain [113]. Celik et al. evaluated 76 young healthy individuals for depression and for latent trigger points of the shoulder girdle muscles [114]. The number of latent trigger points was positively correlated with depression scores. These findings suggest that anxiety, depression, and PTSD may be predisposing factors for MPS.

Several studies addressed the possible mechanisms which allow mental health conditions to play a role in the development of MPS. Torres et al. used transcranial magnetic stimulation (TMS) and quantitative sensory testing (QST) in 47 women with MPS and 11 healthy controls, demonstrating a positive correlation between anxiety and intracranial facilitation at the baseline and after painful stimuli and a negative correlation between anxiety and cortical silent periods [115]. These correlations were only observed among MPS patients. Similar findings were reported by Volz et al. for catastrophizing [116].

The bi-directional association of sleep disturbances and pain, both acute and chronic, is well-documented: sleep complaints are present in 67–88% of chronic pain disorders [117] and at least 50% of individuals with insomnia suffer from chronic pain [118]. Longitudinal population-based studies, as summarized by Finan et al. [119] and Smith et al. [120], indicate that sleep disturbances, most commonly insomnia, may predate the development of chronic regional and widespread pain. Although the association between insomnia and musculoskeletal pain is well documented [121,122,123], we found only one study exploring the association with MPS. In this prospective study, 7895 participants (1579 patients with insomnia and 6316 controls) who were selected from the Taiwan National Health Insurance Research Database were observed for a maximum of 10 years to determine the incidence of newly diagnosed MPS. Individuals with insomnia had a twofold risk of developing MPS [124]. In summary, mental health conditions, specifically anxiety, depression, and PTSD, as well as insomnia, seem to predispose patients to the development of MPS, suggesting that these conditions should be evaluated and treated in individuals with MPS.

## 3. Discussion

The association between MPS and several comorbid medical conditions is reviewed here. While MPS is a common primary cause of pain, evidence is mounting that it may also accompany multiple pathologies, thus contributing to their symptomatic burden. As an example, in a large proportion of individuals suffering from cancer pain, a myofascial component of pain has been identified. Similarly, in patients with neuropathic pain, CRPS, and headaches, MPS is often present. The implications of this observation extend beyond epidemiology and bear clinical pertinence.

### 3.1. Clinical Implications

The existence of MPS that is comorbid with other pathologies holds both diagnostic and therapeutic implications. Diagnostically, it is important to realize that MPS can coexist with other medical conditions. Clinicians new to the field of musculoskeletal medicine readily recognize MPS in a very high proportion of patients presenting with various pain conditions. As an example, we have previously reported that 82% of patients seen at a secondary pain clinic setting were diagnosed with MPS, many of whom had other comorbid pain conditions [20]. It is thus important for clinicians to acknowledge that reaching a diagnosis of MPS does not end the diagnostic process and that comorbid MPS remains a possibility to be considered, such that potential underlying diagnoses are not missed. While we might expect most cases of MPS seen in primary care to be primary, the possibility of comorbid MPS should be kept in mind, especially in individuals with an unexpected presentation of pain or with an atypical response to treatment.

Therapeutically, comorbid MPS holds several implications: firstly, MPS could be an unrecognized source of pain in patients with a variety of medical conditions. In these patients, treatment of the myofascial component of the pain may result in significant improvement of their symptomatic burden. This has been demonstrated in a number of cancer pain settings, as well as in osteoarthritis, CRPS, pancreatitis, and other pathologies. The recognition and treatment of comorbid MPS may thus prove highly valuable in a clinical setting. Secondly, comorbid MPS may be more resistant to treatment than primary MPS, as the comorbid medical condition may serve as a perpetuating factor, irritating the muscles and preventing resolution. These concepts have been demonstrated in patients with painful osteoarthritis of the knee, where treatment of the myofascial component led to a clinically meaningful but short-lived effect [45,48,49]. Finally, when treating a patient with a pain syndrome accompanied by comorbid MPS, clinicians are sometimes faced with a dilemma: does the patient’s pain stem from poor control of the primary pain condition or from the contribution of a putative myofascial component? The answer dictates whether specific treatment for the primary condition should be offered or rather treatment for the myofascial component. A similar problem has been described in the case of comorbid fibromyalgia, which often accompanies inflammatory rheumatic diseases [125]. In this case, the recognition of comorbid fibromyalgia may obviate the need to escalate the anti-inflammatory treatment. Similarly, proper recognition and treatment of comorbid MPS may prevent the need for escalation of treatment modalities aimed at better control of the primary condition.

### 3.2. Comorbid Versus Secondary MPS

Comorbid MPS and secondary MPS are sometimes used interchangeably in the literature to describe MPS that occurs in the presence of, and, possibly, as a result of, another medical condition [1]. Nevertheless, the meanings of these expressions differ: secondary MPS implies a causal relationship between the primary medical condition and MPS, such that MPS is triggered by the primary pathology. Comorbid MPS signifies an association between the prevalence of the two syndromes, not implying a causal correlation in one direction or another. While we find it plausible that comorbid MPS is often secondary (i.e., resulting from the primary condition), sufficient evidence to support this hypothesis is lacking. For this reason, the term comorbid myofascial pain syndrome was preferred here.

When considering MPS that is comorbid with another condition, several possible associations may be contemplated. (1) MPS may predate the comorbid condition. In this case, the condition could develop irrespective of a pre-existing MPS, or alternatively MPS could trigger the development of the comorbid condition. The latter association has been suggested for knee osteoarthritis, whereby soft-tissue changes have been proposed to predate the onset of joint changes [126]. It is important to note that the development of MPS concomitantly with a comorbid entity can and will dictate the treatment of both. (2) MPS may be triggered by the comorbid condition. This association may be plausible in MPS that is comorbid with cancer or infection but, as stated above, sufficient evidence establishing causation is lacking. (3) Both MPS and other pain conditions may develop independently in a person, possibly as a result of higher susceptibility to pain. One such example is the higher prevalence of MPS in individuals with fibromyalgia and other nociplastic disorders [127,128]. As the diagnosis of MPS in a patient with chronic widespread pain is challenging and controversial, we have decided to exclude MPS that is comorbid with nociplastic conditions from this review [127].

### 3.3. Potential Role of Myofascial Trigger Points

The data presented here represent the growing evidence showing that myofascial pain is a common comorbidity associated with multiple painful and non-painful medical conditions. We acknowledge that our interpretation of the data, i.e., that the increased frequency of MPS represents comorbid MPS, is only one possible point of view. An alternative interpretation could be that various pain conditions result in the appearance or activation of trigger points. In this case, MTrP would be an epiphenomenon of any pain condition.

When reviewing the literature, the terms myofascial pain syndrome and myofascial trigger points are sometimes used interchangeably, to the extent that they are almost enmeshed. It is unclear and a subject of debate whether the presence of MTrP defines MPS, accompanies it, or is possibly an epiphenomenon of the syndrome. As stated in the introduction, this review attempts to circumvent the debate surrounding the role of MTRrP in MPS. Thus, we focus on the increased prevalence of myofascial pain in other medical conditions as a clinical phenomenon while leaving the mechanistic interpretations to the readers. We suggest that the recognition of comorbid myofascial pain is pertinent to all clinicians, regardless of their stance in the trigger point controversy.

### 3.4. Limitations

When considering the prevalence of MPS in patients with other medical conditions, there are several limitations to consider. Firstly, the diagnostic criteria of MPS are a topic of continued debate, leading to the use of different diagnostic criteria in different studies. With the advent of the international consensus criteria for the diagnosis of myofascial trigger points, one can hope for better standardization in future studies [12]. Secondly, the lack of consistent diagnostic criteria leads to a wide variety in the estimated prevalence of MPS in the general population [18,20,129]. Thirdly, the same problem leads to an inconsistent estimation of the prevalence of MPS in patients with comorbid medical conditions. Taken together, the ability to compare the prevalence of MPS that is comorbid with certain pathologies with that of the general population is hampered by the lack of methodologically sound data.

## 4. Conclusions

In conclusion, MPS seems to be a common syndrome accompanying multiple medical conditions, negatively affecting the quality of lives of affected patients. The recognition of comorbid MPS may prevent missing critical diagnoses and allow for treatment directed at the MPS component of patients’ pain. We believe that the concept of comorbid MPS should be included in the syllabus of MPS education and considered by clinicians treating individuals with pain.

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
