# Peer review of "The Case for Comorbid Myofascial Pain—A Qualitative Review"

_ijerph, 2020, doi:10.3390/ijerph17145188_

Round 1

Reviewer 1 Report

The manuscript is very interesting and raises very important aspects of MPS that are important for the physician when treating patients with pain. I am very happy to read an article about MPS, in the current situation when many authors question the existence of myofascial pain and trigger points. I found in the text only a few minor errors, they were marked in the attached file. I find the work very valuable and I support admission to print in its current form after the changes have been made

Author Response

We appreciate these positive comments. All suggestions were accepted and the text was revised accordingly. Of note – reviewer 1 suggested changing comorbid to co-morbid. We could find references for both spelling options and thus kindly ask the editor to help us choose the more appropriate option.

Reviewer 2 Report

I found the article very reach and interesting. However, in my opinion it cannot be published in current form.

MTrPs may be divided into active and latent. Active MTrPs cause pain spontaneously and may limit the range of motion, and impair muscle strength. Whereas latent MTrPs are characterized mainly with pain reproducible by palpation. Latent MTrPs may become active ones and reproduce various symptoms. This is important clinical implication. While reading the paper it is unclear which type of MTrPs was considered during the analysis. In my opinion it is a key factor which should be included in paper. If latent MTrPs are coexisting with other medical conditions they are clinically irrelevant. Active MTrPs with wide range of symptoms should be the objects of interest.

Diagnosis of MTrPs is based mainly on clinical examination. Nevertheless, available reports provide data confirming usage of ultrasound elastography, infrared thermography and myotone in diagnosis of MTrPs. As those tools are not gold standard they may be helpful. I think such information should be included in paper.

The diagnosis of MTrPs may be confirmed with symptoms reproduction during examination. If symptoms reported by the patient cannot be provoked during examination it may indicate for other medical conditions. This correlates with types of MTrPs.

Indeed, MTrPs can coexist with other medical conditions. However, it is a key-factor to conduct the differential diagnosis. Most of medical professionals, in particular physiotherapists, are conscious of coexisting conditions during the diagnosis thus such information do not shed new light into MTrPs diagnosis.

Author Response

We appreciate these comments. The INTRODUCTION and DISCUSSION were revised to discuss the potential role of myofascial trigger points. Regarding the suggested elaboration of imaging technologies assisting the diagnosis of MFP, we acknowledge that the field is rapidly evolving, however we find that discussion on new diagnostic technologies for MPS fall beyond the scope of this review. We agree that most experienced clinicians are capable of elaborating a differential diagnosis for pain in various body regions, however we find it useful to draw the attention of clinicians to the possibility that myofascial pain can accompany other medical conditions on one hand, and that the treatment of myofascial pain may relieve some of the symptoms of various comorbidities.

Reviewer 3 Report

This paper brings together what others have already described in a concise and organized manner and as such, you are complimented on taking the time to prepare this manuscript. I would like you to consider whether the notion of a comorbid "myofascial pain syndrome" may be less likely if one were to accept that one of the characterizing features of persistent pain is the presence of trigger points and their associated features such as local and referred pain. In other words, instead of suggesting that there is such a thing as comorbid MPS, perhaps the focus should be on the presence of TrPs in most, if not all, persistent pain states.

I believe that the paper would improve if the authors would contemplate and discuss this and other possible and relevant options. I do not know whether you have considered that potentially, trigger points may be a feature of pain and pain conditions and perhaps rejected that idea. But, it seems conceivable that when patients present with pain - irrelevant of other diagnoses – perhaps the presence of trigger points should be expected rather than the presence of myofascial pain syndrome. The authors opted to consider myofascial pain syndrome as the co-morbid diagnosis, but is there any evidence that this really is a syndrome if one would consider that trigger point may be an epiphonema to pain? Personally, this reviewer is not convinced that the issue is that MPS is a comorbid diagnosis or condition if indeed trigger points are the missing characteristic in descriptions of pain, and especially persistent pain. Perhaps this is a different discussion all together, but the notion of a comorbid myofascial pain syndrome is much less appealing to me than the recognition that one of the characterizing features of persistent pain is the presence of trigger points, and their associated features such as local and referred pain.

From my point of view, including this discussion both in the introduction and discussion sections would make this a much more valuable paper.

I made a few less important observations:

Reference 3 is not the most recent edition of this publication. Any reason why you are not citing the most current edition? Donnelly, J 2019 Travell, Simons & Simons’ Myofascial Pain and Dysfunction: The Trigger Point Manual. Baltimore, Wolters Kluwer.

Introduction line 27 and line 29: try to avoid using the exact same phrase of “widely accepted” this closely together in the paper.

It is somewhat surprising that the authors did not include the studies by Jarrell et al. and Montenegro et al. They are not essential, but do support the main point of the paper.

Jarrell, J, 2004. Myofascial dysfunction in the pelvis. Current pain and headache reports, 8, 452-456.

Jarrell, J, 2011. Endometriosis and abdominal myofascial pain in adults and adolescents. Current Pain and Headache Reports, 15, 368-376.

Jarrell, J, Giamberardino, MA, Robert, M & Nasr-Esfahani, M, 2011. Bedside testing for chronic pelvic pain: discriminating visceral from somatic pain. Pain Res Treat, 2011, 692102.

Jarrell, JF, Vilos, GA, Allaire, C, Burgess, S, Fortin, C, Gerwin, R, Lapensee, L, Lea, RH, Leyland, NA, Martyn, P, Shenassa, H, Taenzer, P & Abu-Rafea, B, 2005. Consensus guidelines for the management of chronic pelvic pain. J Obstet Gynaecol Can, 27, 781-826

Montenegro, ML, Gomide, LB, Mateus-Vasconcelos, EL, Rosa-E-Silva, JC, Candido-Dos-Reis, FJ, Nogueira, AA & Poli-Neto, OB, 2009. Abdominal myofascial pain syndrome must be considered in the differential diagnosis of chronic pelvic pain. European Journal of Obstetrics, Gynecology, and Reproductive Biology, 147, 21-24.

Author Response

We appreciate these positive comments. We completely agree that the results presented in this review can be interpreted in more than one way. Reiterating our response to the comments made by the Editor, the aims of this review were: 1) to provide an overview of the co-occurrence of MPS as a clinical phenomenon with other painful and non-painful medical conditions; and 2) to discuss the possible clinical diagnostic and therapeutic implications of this observation. We deliberately attempted to avoid the discussion on potentially pathophysiologic mechanisms. However, we acknowledge that such a discussion is important here and thank the reviewer for commenting on this. We have added a paragraph to DISCUSSION, outlining the potential role of MTP and alternative interpretation of the results including activation of MTP as an epiphenomenon of other pain conditions as reviewer 3 reasonably suggested.

These  references (suggested in the review)have been added to the manuscript. Thank you.